# Role of Serotonin in the Maintenance of Inflammatory State in Crohn’s Disease

**DOI:** 10.3390/biomedicines10040765

**Published:** 2022-03-24

**Authors:** Simona Pergolizzi, Alessio Alesci, Antonio Centofanti, Marialuisa Aragona, Socrate Pallio, Ludovico Magaudda, Giuseppina Cutroneo, Eugenia Rita Lauriano

**Affiliations:** 1Department of Chemical, Biological, Pharmaceutical and Environmental Sciences, University of Messina, 98166 Messina, Italy; elauriano@unime.it; 2Department of Biomedical and Dental Sciences and Morphofunctional Images, University of Messina, 98125 Messina, Italy; ludovico.magaudda@unime.it (L.M.); giuseppina.cutroneo@unime.it (G.C.); 3Department of Veterinary Sciences, University of Messina, Polo Universitario dell’Annunziata, 98168 Messina, Italy; mlaragona@unime.it; 4Department of Clinical and Experimental Medicine, University of Messina, 98147 Messina, Italy; socrate.pallio@unime.it

**Keywords:** Crohn’s disease, serotonin, dendritic cells, myofibroblast, goblet cells

## Abstract

Crohn’s disease (CD) is a chronic intestinal inflammation considered to be a major entity of inflammatory bowel diseases (IBDs), affecting different segments of the whole gastrointestinal tract. Peripheral serotonin (5-HT), a bioactive amine predominantly produced by gut enterochromaffin cells (ECs), is crucial in gastrointestinal functions, including motility, sensitivity, secretion, and the inflammatory response. These actions are mediated by a large family of serotonin receptors and specialized serotonin transporter (SERT) located on a variety of cell types in the gut. Several studies indicate that intestinal 5-HT signaling is altered in patients with inflammatory bowel disease. Paraformaldehyde-fixed intestinal tissues, obtained from fifteen patients with Crohn’s disease were analyzed by immunostaining for serotonin, Langerin/CD207, and alpha-Smooth Muscle Actin (α-SMA). As controls, unaffected (normal) intestinal specimens of seven individuals were investigated. This study aimed to show the expression of serotonin in dendritic cells (DCs) and myofibroblast which have been characterized with Langerin/CD207 and α-SMA, respectively; furthermore, for the first time, we have found the presence of serotonin in goblet cells. Our results show the correlation between different types of intestinal cells in the maintenance of the inflammatory state in CD linked to the recall of myofibroblasts.

## 1. Introduction

Crohn’s disease (CD) is a chronic recurrent inflammatory bowel disease that affects millions of people around the world [1]. Approximately one-third of patients with Crohn’s disease display a distinct fibrostenosing phenotype which predisposes them to recurrent intestinal stricture formation, most commonly involving the small bowel [2]. Symptoms are mainly represented by diarrhea, abdominal pain, and rectal bleeding [3]. Intestinal fibrosis, commonly defined as an excessive deposition of extracellular matrix (ECM) resulting from chronic inflammation and impairment of intestinal wound healing, represents a serious complication of Intestinal Bowel Diseases (IBDs) and has important clinical implications [4]. In ulcerative colitis (UC), the involvement of the mucosal and submucosal layers causes a thickening of the muscularis mucosae with an accumulation of ECM that may contribute to shortening or stiffening of the colon, whereas in CD, the transmural nature of the inflammatory process is followed by bowel wall thickening, and eventually the formation of stenosis [5].

In Crohn’s disease, inflammation involves intestinal mesenchymal cells that are in a constant state of trans- and dedifferentiation among fibroblast, myofibroblast, and smooth muscle cell phenotypes (SMCs) in a little-known interaction with immune cells and environmental stimuli [6,7]. The intestinal fibrosis initiation and propagation foresees two main parallel events, which are the expansion of the smooth muscle layers and the fibrosing process [6,7]. Moreover, SMCs, project out from muscularis mucosae and muscularis propria into the fibrotic submucosa [8]. Studies on Crohn’s disease have shown that, during stricture formation, α-Smooth Muscle Actin (α-SMA) stains the submucosa [9] and subserosa [10], demonstrating that there is an accumulation of myofibroblasts in layers most affected by fibrosis [6]. In the gut, fibroblasts can stimulate their migration through autocrine or paracrine processes [11,12].

Serotonin (5-hydroxytryptamine, 5-HT) is a highly preserved and ubiquitous endogenous monoamine signaling molecule [13] regulating a variety of biological processes such as sleep, appetite, and mood; and it is a vasoactive amine playing a central role in many organ systems such as in intestinal motor and secretory function [14,15,16]. Moreover, 5-HT has significant effects on inflammation and immunity [17]. E. Tarbit et al., (2021) demonstrated the presence of 5-HT1A, 5-HT2A, and 5-HT2B (serotonin receptors) in adult rat cardiac fibroblasts and myofibroblast cells, suggesting that serotonin levels may contribute to the pathogenesis of heart failure via serotonin receptors [18]. Anna Löfdahl et al., (2020) in their review article state the contribution of serotonergic signaling in pulmonary fibrosis [19]. Most of the peripheral serotonin is synthesized by precursor L-tryptophan in the enterochromaffin cells (ECs) of the intestine, secreted into the bloodstream, and then taken up by circulating platelets [20,21,22]. 5-HT also regulates epithelial proliferation and turnover of the intestinal mucosal epithelium [23,24]. A very important finding for our study is that intestinal serotonin, once released, can activate receptors located in various cell types including goblet cells [25]. 

The imbalance between proinflammatory and anti-inflammatory cytokines and chemokines plays a critical role in the differentiation, activation, and migration of scar-producing myofibroblasts during the development of fibrosis [7]. 

In our previous work, we have shown that dendritic cells (DCs), in ulcerative colitis, are involved in the synthesis of neurotransmitters [16]. Dendritic cells are antigen-presenting cells located in contact with the environment (skin and mucous membranes) that recognize and process antigens for presentation to T cells [26,27,28,29]. DCs are considered “bridging” cells between innate and acquired immunity and are involved in immune response [30,31]. Intestinal DCs are the major source of proinflammatory mediators, including cytokines, and play an important role in the induction and maintenance of chronic inflammation in IBD [32,33,34].

Langerin/CD207 is a c-type lectin expressed by different types of DCs: skin’s Langerhans cells, dendritic cells present in the cornea and ocular surface [35], and by a subset of dendritic cells present in most connective tissues, including the dermis, lung, kidney, and liver [36]. Furthermore, DCs that represent the main population of antigen-presenting cells in gut-associated lymphoid tissues and lamina propria of mice and rat models as well as in humans [37] are Langerin/CD207 positive [38].

In the present study, we have shown the expression of serotonin in myofibroblast and dendritic cells in colocalization with α-SMA and Langerin/CD207, respectively; furthermore, we have found for the first time the presence of serotonin in goblet cells.

Our results show the correlation between different types of intestinal cells such as diffuse endocrine system cells, immune cells, and goblet cells in the maintenance of the inflammatory state in CD linked to the accumulation of myofibroblasts during stricture formation.

## 2. Materials and Methods

### 2.1. Samples and Tissue Treatment

Biopsies of inflamed ileum from CD patients and non-inflamed ileum tissue from control patients were collected through ileocolonoscopy. Examined subjects were 22 individuals, 15 patients with a diagnosis of ileal CD, and 7 controls undergoing screening for family history of Crohn’s disease or inflammatory bowel diseases; the mean age of subjects was 21.5 years (range 18–25 years). Patients with CD were categorized based on the presence of blood or mucus in their stool, or both; painful incontinence; nocturnal diarrhea; recent intestinal infections; and they were reviewed again six months after the initial diagnosis. The diagnosis of CD was based on clinical, laboratory, microbial, and endoscopic findings. Diagnostic methodologies are summarized in Table 1. The research was done out in line with the Helsinki Declaration.

All samples were taken at the University Policlinic (Messina, Italy), all patients gave their informed consent, and the study was approved by the ethical council under the number C.E. prot. 103/19. Each sample was immediately fixed in 4% paraformaldehyde in phosphate-buffered saline (PBS), 0.1 mol/L (pH 7.4) for 2–4 h, after removal; then dehydrated in graded ethanol, cleared in xylene, embedded in Paraplast™ (McCormick Scientific, St. Louis, MO, USA), and cut into 5 µm sections. Some sections were deparaffinized and rehydrated, washed in distilled water, and stained with hematoxylin and eosin (H&E) (Carazzi’s Hematoxylin Nuclear staining, 05-M06012; Eosin Y 1% aqueous solution cyto- plasmic staining, 05-M10002, Bio-Optica Milano S.p.a., Milan, Italy) [39,40], and Alcian Blue pH 2.5 Periodic Acid Schiff (AB/PAS) (04-163802, Bio-Optica Milano S.p.a., Milan, Italy) [41,42]. Sections were examined under a Light microscope Eclipse Ci-L (Nikon Corporation, Tokyo, Japan). The micrographs were acquired with a Nikon DSRi2 Camera, equipped with a NIS-Element F Software.

### 2.2. Immunofluorescence

The serial sections were deparaffinized and rehydrated, rinsed several times in PBS, and incubated in 0.3% H_2_O_2_ in PBS solution for 3 min to prevent the activity of endogenous peroxidase, finally blocked in 2.5% bovine serum albumin (BSA) for 1 h. Polyclonal anti serotonin (5-HT) was used in double-label experiments with a monoclonal antibody anti-Langerin (Table 2) and, with a monoclonal antibody anti-α-Smooth Muscle Actin (Table 2), as previously described for a wide variety of tissues [35,36,41,42,43,44], with an overnight incubation at 4 °C in a humid chamber. After repeated rinsing in PBS, the sections were incubated for 1 h with secondary antisera: Alexa Fluor 488 donkey anti-mouse IgG FITC conjugated, and Alexa Fluor 594 donkey anti-rabbit IgG TRITC conjugated (Table 2), respectively. After washing, the slices were covered slid, and mounted with Fluoromount^TM^ Aqueous Mounting Medium (F4680 Sigma-Aldrich, St. Louis, MO, USA) to avoid photobleaching. Control experiments excluding primary antibodies were performed (data not shown).

### 2.3. Laser Confocal Immunofluorescence

Sections were analyzed and images acquired using a Zeiss LSM DUO confocal laser scanning microscope with META module (Carl Zeiss MicroImaging GmbH, Jena, Germany) equipped with an argon laser (458, 488 l) and 2 helium-neon lasers (543 and 633 l). All pictures were digitalized at an 8-bit resolution into a 2048 × 2048-pixel array. Optical slices of fluorescence samples were acquired using a 1-min, 2-s scanning speed and up to 8 averages using a helium-neon laser (543 nm) and an argon laser (458 nm). We obtained 1.50-µm-thick sections using a pinhole of 250; the images captured were processed using Zen 2011 (LSM 700 Zeiss software). To reduce photodegradation, each image was captured quickly. Adobe Photoshop CC (Adobe Systems, San Jose, CA, USA) was used to crop digital pictures and create the figure montage. The “display profile” function of the laser scanning microscope was used to show the intensity profile on an image along a freely selectable line. The intensity curves are shown in the graphs next to the scanned images.

### 2.4. Statistical Analysis

To collect data for statistical analysis, five sections and ten fields were inspected for each subject, respectively, for patients with Crohn’s disease and control subjects. The fields were chosen subjectively depending on the positivity of the cells. Each field was examined using ImageJ software [45]. A “Threshold” filter and a mask were used to identify cells and eliminate the background after converting the image to 8 bits. The “Analyze particles” plug-in was then used to count the cells. The statistical significance of the number of DCs Langerin positive, goblet cells 5-HT positive, and myofibroblast α-SMA positive was investigated using ANOVA. The statistical analysis was carried out using SigmaPlot version 14.0 (Systat Software, San Jose, CA, USA). The data were presented as mean values with standard deviations (Δs). *p* values less than 0.05 were considered statistically significant in this sequence: ** *p* ≤ 0.01, * *p* ≤ 0.05.

## 3. Results

### 3.1. Light Microscopy

The inner surface of the small intestine is characterized by the presence of a large number of protrusions (folds, villi, microvilli) to perform the task of absorbing nutrients. In the healthy intestine, the villi are covered by a single-layer columnar epithelium, consisting mainly of absorbent cells, the enterocytes. In addition to enterocytes, there are three different cell types with specific secretory functions, namely goblet cells (mucus constituents), enteroendocrine cells (peptide hormones), and Paneth cells (antimicrobial active substances). The observed sections of the healthy ileal mucosa stained with H&E showed a regular intestinal epithelium with columnar cells lining the villi towards the luminal surface, mainly comprising enterocytes, with the typical brush border, and goblet cells with rounded calyxes. The lamina propria consisted of stromal elements, arranged regularly, and showed no fibrotic, proliferative, or inflammatory changes. Goblet cells can be highlighted with Alcian Blue/Periodic Acid Schiff (AB/PAS) staining showing different colors depending on the composition of the secreted mucus. Goblet cells containing glycoconjugate acid are Alcian Blue (AB) positive, neutral mucins are stained in magenta with PAS, and finally, mucus that has both types of mucopolysaccharides is purple colored, AB/PAS. In the control sections, goblet cells secreting acidic and neutral mucin were highlighted in purple with AB/PAS staining. The ileal sections of the CD samples, stained with H/E, showed an altered epithelial arrangement, with enterocytes constricted between the goblet cells with the elongated calyx. In the lamina propria, it was possible to observe an accumulation of mesenchymal cells immersed in an abundant collagen matrix and dilated blood vessels. AB/PAS-treated histological sections of the CD-affected intestine showed blue-stained goblet cells, indicating acid mucus production (Figure 1).

### 3.2. Double Labeling of 5-HT and α-SMA

In control samples, we detected enteroendocrine cells (ECs) 5-HT positive (red fluorescence) in the epithelial compartment (arrows). A surprising finding concerns goblet cells, in whose cytoplasm we observed positivity to 5-HT (asterisks) (Figure 2). Green fluorescence for *α*-SMA was detected in lamina propria stromal cells (large arrows) (Figure 2). In these samples, some stromal cells were double labeled (yellow fluorescence) with the two tested antibodies (large arrows) (Figure 3).

In CD samples, goblet cells’ cytoplasm was not labeled with 5-HT (asterisks) (Figure 2). In the lamina propria of CD samples, numerous α-SMA-positive cells (subepithelial myofibroblasts) were detected (Figure 2), which colocalized with 5-HT (large arrow) (Figure 3). A cluster of subepithelial myofibroblasts was found in the lamina propria, which colocalized markedly with 5-HT and α-SMA (large arrows) (Figure 3). Statistical analysis demonstrated an increase in myofibroblasts α-SMA/5HT positive in CD (Table 3). To confirm the protein staining patterns, we used the “display profile” software function of the laser scanning microscope for selected samples. This additional analysis, which reveals the fluorescence intensity profile on an image along a freely selectable line, converted the immunofluorescent signal into a graph. The display profile of the control samples showed clear fluorescence peaks for α-SMA and 5-HT distant from each other, while peak overlap was observed in the CD samples.

### 3.3. Double Labeling of 5-HT and Langerin/CD207

In the control samples, 5-HT positive enteroendocrine cells (red fluorescence) were observed among epithelial cells (arrows). Additionally, in these samples, the positivity of the goblet cell cytoplasm to 5-HT was evident (asterisks), and higher than in CD (Figure 4). Langerin-positive dendritic cells (green fluorescence) were observed in the thickness of the epithelium and lamina propria (double arrows) (Figure 4); these cells showed colocalization with 5-HT (yellow fluorescence) (Figure 5).

In CD gut specimens, 5-HT positivity in goblet cells was absent (asterisks); 5-HT positive enteroendocrine cells were highlighted (arrows) (Figure 4). Langerin positive dendritic cells (green fluorescence) were located at the apex of the villi and in the lamina propria below the epithelium (double arrows) (Figure 4). In the lamina propria, abundant dendritic cells colocalized with Langerin/CD207 and 5-HT (Figure 5). The “display profile” software function of the laser scanning microscope for selected samples was used. The display profile showed overlapping fluorescence peaks for Langerin and 5-HT in both the control and CD samples.

## 4. Discussion

Crohn’s disease is a recurrent systemic inflammatory disease that mostly affects the gastrointestinal tract but can also cause extraintestinal symptoms and immunological abnormalities. 

The first line of defense of the mucosal immune system is a single polarized layer of epithelial cells covered with mucus biofilms secreted by goblet cells with the function of a chemical-physical barrier [46]. Possible dysfunction of this barrier increases the transfer of substances, including bacteria, into the lamina propria, leading to chronic inflammation of the intestine [47]. A study by Yamada et al., (2019) noted that levels of mucin related to its O-glycans were higher in patients with intestinal inflammation than in healthy subjects [48]. Overproduction of mucins and abnormal glycosylation are typical of Crohn’s disease. Dorofeyev et al., (2013) showed a prevalence of sulfated mucins (acid mucins) in patients with CD [49]. Abnormalities in Paneth’s cells and neuroendocrine cells are related to changes in microbiota and secreted mucus [50]. These changes can alter the viscoelastic properties of the mucus, affecting the inflammatory state [51]. In this study, AB/PAS demonstrated the presence of glycoconjugate acid in gut goblet cells of CD patients sections, according to previous studies [49,51].

Peripheral 5-HT is a bioactive amine predominantly produced by enterochromaffin cells (ECs) of the gastrointestinal (GI) tract [52]. Engevik et al., (2019) showed that commensal microbes can excite 5-HT secretion by ECs activating serotonin receptor 4 (5-HTR4) on goblet cells to endorse MUC2 excretion [25]. In this study, we demonstrated, for the first time, the presence of 5-HT in goblet cells cytoplasm of control samples, whereas the positivity to serotonin was few or absent in CD patients’ epithelial goblet cells. In agreement with Roberto Chiocchetti (2022), we hypothesize that the presence of serotonin in the cytoplasm of goblet cells may be due to specialized serotonin transporter (SERT) present in the goblet cells [53]. SERT is expressed by a variety of mucosal immune cells, including macrophages, mast cells, lymphocytes, dendritic cells [54], and intestinal epithelial cells, such as enterocytes [13,55,56,57] and goblet cells [53]. Serotonin transporter phosphorylated is internalized and its expression can be altered in inflammatory bowel disease (IBD) [13,57,58,59,60].

Jorandli et al., (2020) demonstrated that reduction of serotonin reuptake transporter (SERT or 5-HTT) observed in active CD and UC may contribute to the increased interstitial serotonin level associated with intestinal inflammation [13]; we hypothesize that in this way, a large amount of serotonin can contact the dendritic cells and myofibroblasts present in the lamina propria.

Changes in enteroendocrine cell number and secretion have been seen in both murine and human studies during inflammation [61,62], and an array of enteroendocrine cell peptide receptors has been found in the immune system that serves the gut [63]. Immune cells have a surprising number of vesicular neurotransmitter receptors and transporters for the hormone peptides produced by enteroendocrine cells [54,63]; in our previous research, we found that UC dendritic cells contain 5-HT and the vesicular acetylcholine transporter (VAChT) [16], demonstrating that bi-directional signaling in the immunoendocrine axis has exciting potential. Intestinal antigen-presenting dendritic cells (DCs) play a critical role in the initiation and maintenance of immune responses [64,65]. Based on their cell-surface phenotypic and functional features, distinct subgroups of DCs have been identified in mice and humans. Myeloid and plasmacytoid DCs are two types of DCs seen in humans. DCs may play a role in promoting intestinal inflammation, according to studies in animal models of colitis and human inflammatory bowel disease (IBD) [66,67]. In the present study, DCs co-expressing Langerin/CD207, and 5-HT were detected in villi and lamina propria of CD patients, compared to normal ilea in which Langerin-positive cells located at the villus apex were observed. This result is in agreement with literature data, supporting the hypothesis that positive Langerin cells can migrate from the villi to the lamina propria carrying neurotransmitters such as serotonin. 

Subepithelial myofibroblasts (SEMFs) appear to be key players in the fibrogenesis process, due to the production of ECM components [68]. The subepithelial myofibroblasts, which reside beneath the epithelial barrier in the lamina propria, are involved in fibrosis [69]. Specific cellular markers, such as vimentin and α-smooth muscle actin (α-SMA), have been used to identify them [70]. In case of injury, SEMFs are activated by assuming contractility expressing an excess of α-SMA [35,71,72]. Subsequently, subepithelial myofibroblasts proliferate and move to the wound site, secreting vast amounts of ECM components like collagen and fibronectin to build a temporary barrier to the lumen [70,73]. In pathological conditions like CD, chronic inflammation causes a continual state of activation and proliferation of SEMFs, leading to an imbalance between ECM accumulation and breakdown, and eventually fibrosis [74]. Profibrotic cascades are thought to be initiated by inflammatory reactions originating from either innate or adaptive immunity, according to growing evidence [75]. SEMFs represent a second line of defense and are involved in innate immune responses via TLR expression [76,77,78,79,80,81,82,83]. SEMFs in CD and UC are directly linked to intestinal inflammation through the production of cytokine receptors, according to Filidou et al., (2018), and adaptive immune responses may play a substantial role in fibrogenesis [69]. In this study, we demonstrated the colocalization of serotonin (5-HT) and α-SMA in subepithelial myofibroblasts of CD ileal samples, while in control samples the two antibodies were poorly overlapped.

IBD pathophysiology has long been linked to epithelial barrier disruption and intestinal leakage [84]. However, it has recently been discovered that the epithelium has a role in many other aspects of IBD pathogenesis, including immune control, regeneration, and interactions with the microbiome [85].

Taken together, these findings suggest that serotonin (5-HT), a bioactive amine produced by enterochromaffin cells (ECs) of the gastrointestinal (GI) tract and captured by goblet cells under normal conditions, may play a crucial role in maintaining the CD pathology, through the recruitment and promotion of subepithelial myofibroblasts. Moreover, a higher amount of DCs colocalized with Langerin and 5-HT in Crohn’s disease patients supports the already expressed thesis of the close link between DCs and neurotransmitters.

## Figures and Tables

**Figure 1 biomedicines-10-00765-f001:**
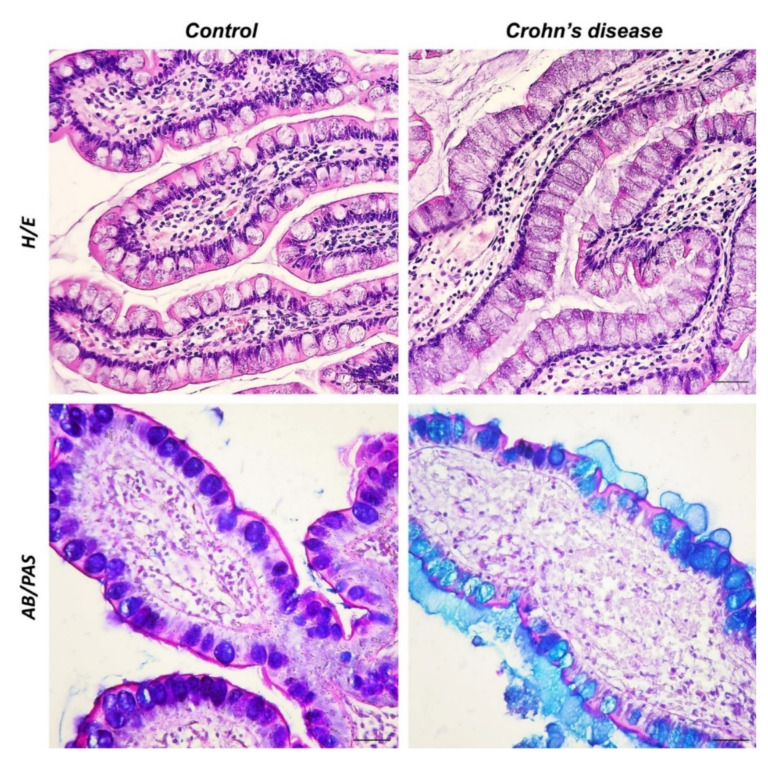
H/E and AB/PAS staining. 40×, scale bar: 40 μm. In the control samples, the intestinal epithelium appeared regular with columnar cells along the villi covering the luminal surface, consisting mainly of enterocytes, and goblet cells with rounded calyxes. The altered epithelial architecture was observed in the CD sections. Rarefied enterocytes were present among goblet cells with an elongated calyx. Infiltration of inflammatory cells, collagen deposition, and accumulation of mesenchymal cells were observed in the lamina propria of CD samples. The control samples stained with AB/PAS showed purple goblet cells, while in the CD samples, the goblet cells were highlighted in blue, with an evident layer of acidophilic mucus.

**Figure 2 biomedicines-10-00765-f002:**
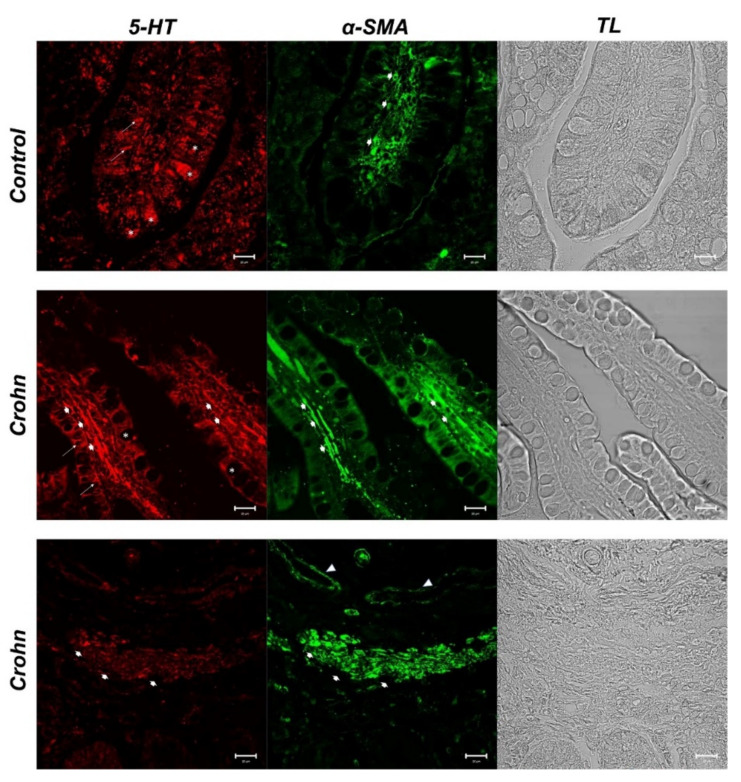
Human intestine. 5-HT and α-SMA, 20×, scale bar 20 nm. In healthy intestine sections, red fluorescence evidenced serotonergic enteroendocrine cells (arrows) and the goblet cell cytoplasm (*). The positivity to α-SMA was observed in stromal cells (fibroblasts and mesenchymal cells) (large arrows). In CD samples, enterochromaffin cells (ECs), positive to 5-HT, were evident in the epithelium (arrows). Lamina propria cells with a high positivity to 5-HT were detectable (big arrows). The goblet cell cytoplasm was 5-HT negative (*). α-SMA positive cells in the lamina propria below the epithelium (large arrows). Blood vessels (arrowhead). The micrographs are also equipped with transmitted light (TL) to visualize the organ morphology.

**Figure 3 biomedicines-10-00765-f003:**
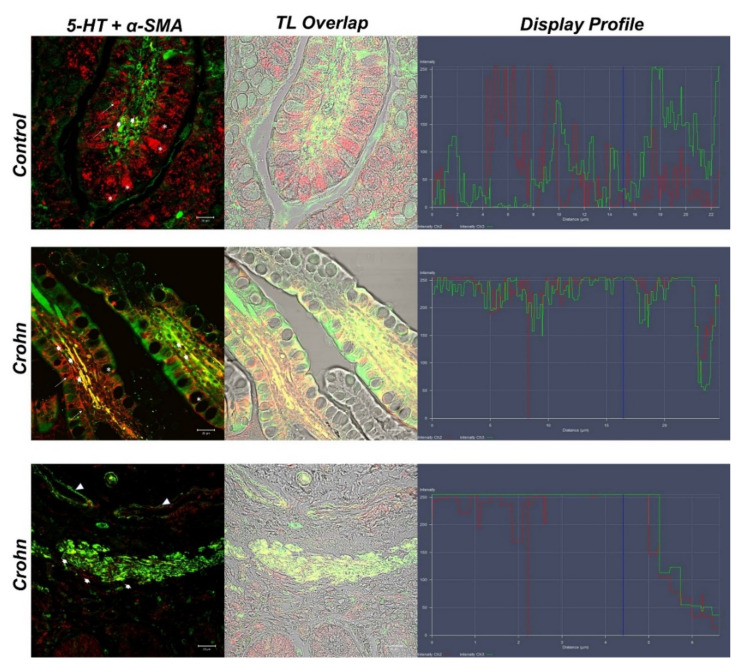
Human intestine, 5-HT/α-SMA colocalization, 20×, scale bar 20 nm. Merge photomicrographs indicate the double localization of two antibodies in myofibroblasts, in which it is evident that the overlap (yellow fluorescence) concerns the CD samples rather than in the healthy intestine. In the control section 5-HT highlighted the goblet cells cytoplasm (*). A cluster of subepithelial myofibroblasts in the lamina propria colocalized with 5-HT and α-SMA (large arrows), blood vessels (arrowheads). The graph represents the “display profile” function of the laser scanning microscope to show the intensity profile of detected antibodies.

**Figure 4 biomedicines-10-00765-f004:**
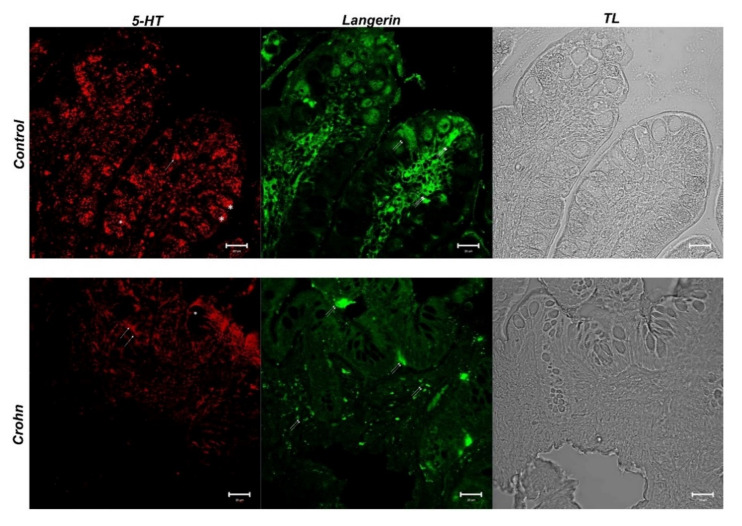
Human intestine. 5-HT and Langerin/CD207, 20×, scale bar 20 nm. In healthy intestine sections, ECs that were serotonin positive between epithelial cells (arrow) were highlighted with red fluorescence. 5-HT positivity was observed in the goblet cell cytoplasm (*). DCs that were Langerin positive (green fluorescence) were shown in the thickness of the epithelium and the lamina propria (double arrow). In CD samples, 5-HT positivity in goblet cells disappeared and it was more evident in stromal cells (arrows). DCs in the epithelium and lamina propria (double arrows) were marked with Langerin in green fluorescence. The micrographs are also equipped with transmitted light (TL) to visualize the organ morphology.

**Figure 5 biomedicines-10-00765-f005:**
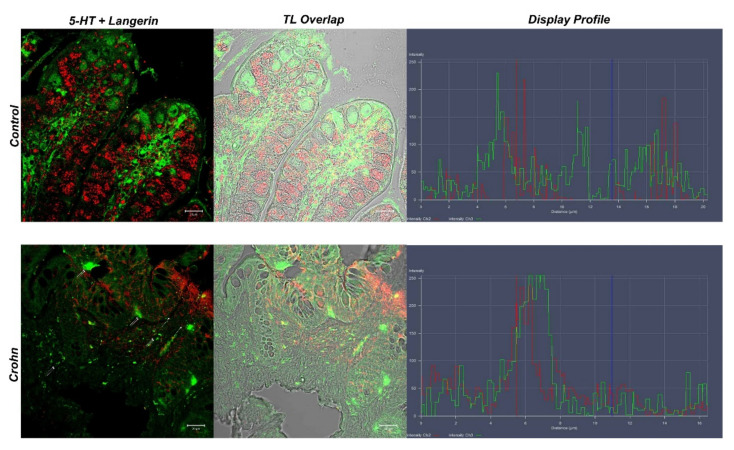
Human intestine. 5-HT/Langerin DC/207 colocalization, 20×, scale bar 20 nm. The merged photomicrograph indicates the double localization of two antibodies in DCs. In the healthy intestine, 5-HT and Langerin showed overlap, which was more evident in CD samples. Stromal cells are highlighted with arrows. The graph represents the “display profile” function of the laser scanning microscope to show the intensity profile of detected antibodies.

**Table 1 biomedicines-10-00765-t001:** Summary of diagnostic studies.

Laboratory studies	Electrolytes, creatinine, erythrocyte sedimentation rate, complete blood count with differential, blood urea nitrogen, liver function tests, transferrin, bilirubin, ferritin, folic acid, vitamin B12, urine strip, C-reactive protein, faecal calprotectin.
Microbial studies	Stool cultures
Endoscopy	Ileocolonoscopy (spared rectum, ileal inflammation, skip lesions, cobble stoning, fissural and longitudinal ulcers, strictures, fistulas). Oesophagogastroduodenoscopy with biopsies when symptoms occur in the upper gastrointestinal tract.The SES-CD criteria (The Simple Endoscopic Index for Crohn’s Disease) for patients with unoperated colon and ileus, and The Rutgeert’s Score for patients with surgery (with particular reference to anastomotic recurrence) were used in the endoscopic assessment of the severity of patients with Crohn’s disease.

**Table 2 biomedicines-10-00765-t002:** Summary of primary and secondary antibodies.

**Primary Antibodies**	**Supplier**	**Catalogue Number**	**Source**	**Dilution**	**Antibody ID**
Anti-serotonin (5-HT)	Sigma-Aldrich	S5545	rabbit	1:500	AB_477522
Langerin (H-4)	Santa Cruz biotechnology, inc	sc-271272	mouse	1:500	AB_10611518
α-Smooth Muscle Actin	Sigma-Aldrich	A5228	mouse	1:200	AB_262054
**Secondary Antibodies**	**Supplier**	**Catalogue Number**	**Source**	**Dilution**	**Antibody ID**
Alexa Fluor 488 anti-mouse IgG FITC conjugated	Invitrogen	A-21202	donkey	1:300	AB_141607
Alexa Fluor 594 anti-rabbit IgG TRITC conjugated	Invitrogen	A32754	donkey	1:300	AB_2762827

**Table 3 biomedicines-10-00765-t003:** Statistical analysis data (±Δs). Mean values of myofibroblasts detected by α-SMA, and of 5-HT positive cells in lamina propria. Myofibroblasts are increased in CD. Statistical analysis of the number of LCs positive to Langerin, and goblet cells detected by 5-HT antibody. LCs are higher in CD.

	Intestine Sections of Patients with Crohn’s Disease	Intestine Sections of Control Subjects
Myofibroblasts α-SMA+	542.38 ± 85.82 **	103.86 ± 9.48 *
5-HT+ cells in lamina propria	369.71 ± 7.76 *	97.13 ± 9.33 *
Merge α-SMA/5-HT	306.75 ± 12.40 *	93.17 ± 9.52 *
LCs Langerin+	816.53 ± 118.91 *	121.63 ± 11.22 **
Goblet Cells 5-HT+	102.38 ± 11.05 **	1056.67 ± 37.14 *
Merge Lan/5-HT	327.52 ± 6.01 **	102.69 ± 11.53 *

** *p* ≤ 0.01; * *p* ≤ 0.05.

## Data Availability

Not applicable.

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
