# Peer review of "Role of Serotonin in the Maintenance of Inflammatory State in Crohn’s Disease"

_biomedicines, 2022, doi:10.3390/biomedicines10040765_

Round 1

Reviewer 1 Report

The article of Simona Pergolizzi et al., entitled “Role of the serotonin in the maintenance of inflammatory state in Crohn’s disease” highlights the significant role of serotonin in Crohn’s disease, suggesting that this bioactive amine implicates in maintaining the CD pathology through the recruitment and promotion of subepithelial myofibroblasts, accumulated in the stricture site. It is a well-structured article that shed some light in the understanding of Crohn’s disease. However, I have a few minor concerns that need to be addressed:

  • In the paragraph 2.1, the authors mention that diagnosis of CD was based on clinical, laboratory, microbial and endoscopic findings. I believe that they should elaborate on the diagnosis of CD by giving more details regarding these findings and, also, incorporate a table presenting these details.
  • The section of results appears to be quite modest. The authors are advised elaborate on their results in order to enrich the particular section.
  • English correction is needed. The manuscript contains grammatical and drafting errors.

Author Response

Dear Referee, thank you for your helpful suggestions. Here are our responses to your comments:

1) A table with diagnostic tests has been added, as requested.

2) The results have been implemented and enriched.

3) The manuscript was corrected by a native English speaker.

Reviewer 2 Report

In the current manuscript “Role of serotonin in the maintenance of inflammatory state in Crohn's disease” authors investigated the serotonin expression in the human control and Crohn’s disease patient’s samples using confocal microscopy.  

Comments:

  1. In figure 1 and 2, 5-HT and α-SMA stain looks gross on intestinal tissues. The intestinal sections need to stain with only secondary antibody as a control to validate specific stain of each.
  2. Co-localization of 5-HT and α-SMA stain need to be quantify and compare in control and Crohn’s disease patient’s samples.
  3. Goblet cell specific stain such as Muc2 can be performed along with 5-HT to confirm authors claim of goblet cell positivity for 5-HT.
  4. To validate 5-HT staining in enteroendocrine cells, chromogranin A and 5-HT colocalization staining should be performed.
  5. Line number 175, scale bar 40 nm is questionable. Please rectify. This is applicable for all figure legends.

Author Response

We thank you for the review, although we do not appreciate the use of some words, which we feel are inappropriate. Below are our responses to the comments received:

1) The confocal microscopy images have been changed and the negative control images, which we had already done and indicated in the previous text with the indication "data not shown", have been included in the panels.

2) To provide additional clarification regarding colocalization, relative graphs, obtained with confocal microscopy, and transmitted light colocalization images have been added.

3) We performed MUC2 for intestinal mucosal cells but cannot include the images as they are used to produce another article. However, our fluorescent marker for MUC2 is red, as is the one for 5-HT, so we could not have implemented colocalization anyway. However, we added images of tissue stained with AB / PAS, mucopolysaccharide specific histochemical staining, thus highlighting not only mucosal cells but also the diversity of mucus composition.

4) Unfortunately, we do not have chromogranin A, but we used transmitted light images, colocalization graphs, and negative controls to highlight the validity of our reactions.

5) The scale bar has been reviewed and corrected in every image.

Round 2

Reviewer 2 Report

Authors addressed all the comments.